# Zeolites as Adsorbents and Photocatalysts for Removal of Dyes from the Aqueous Environment

**DOI:** 10.3390/molecules27196582

**Published:** 2022-10-04

**Authors:** Marina Rakanović, Andrijana Vukojević, Maria M. Savanović, Stevan Armaković, Svetlana Pelemiš, Fatima Živić, Slavica Sladojević, Sanja J. Armaković

**Affiliations:** 1Faculty of Technology, University of Banja Luka, 78000 Banja Luka, Bosnia and Herzegovina; 2University of Novi Sad, Faculty of Sciences, Department of Chemistry, Biochemistry and Environmental Protection, 21000 Novi Sad, Serbia; 3University of Novi Sad, Faculty of Sciences, Department of Physics, 21000 Novi Sad, Serbia; 4Faculty of Technology Zvornik, University of East Sarajevo, 75400 Zvornik, Bosnia and Herzegovina; 5University of Kragujevac, Faculty of Engineering, 34000 Kragujevac, Serbia

**Keywords:** Methylene blue, Rhodamine B, BETA zeolite, ZSM-5 zeolite, NaY zeolite, water purification, molecular modeling, DFT calculation

## Abstract

This study investigated the potential of zeolites (NH_4_BETA, NH_4_ZSM-5, and NaY) to remove two frequently used dyes, methylene blue (MB) and rhodamine B (RB), from an aqueous environment. The removal of dyes with zeolites was performed via two mechanisms: adsorption and photocatalysis. Removal of dyes through adsorption was achieved by studying the Freundlich adsorption isotherms, while photocatalytic removal of dyes was performed under UV irradiation. In both cases, the removal experiments were conducted for 180 min at two temperatures (283 K and 293 K), and dye concentrations were determined spectrophotometrically. Additionally, after photodegradation, mineralization was analyzed as chemical oxygen demand. A computational analysis of the structures of MB and RB was performed to gain a deeper understanding of the obtained results. The computational analysis encompassed density functional theory (DFT) calculations and analysis of two quantum-molecular descriptors addressing the local reactivity of molecules. Experimental results have indicated that the considered zeolites effectively remove both dyes through both mechanisms, especially NH_4_BETA and NH_4_ZSM-5, due to the presence of active acidic centers on the outer and inner surfaces of the zeolite. The lowest efficiency of dye removal was achieved in the presence of NaY zeolite, which has a lower SiO_2_/Al_2_O_3_ ratio. A more efficient reduction was completed for RB dye, which agrees with the computationally obtained information about reactivity.

## 1. Introduction

Clean and unpolluted water is an irreplaceable resource that enables life on Earth. Water quality has deteriorated yearly due to rapid economic and industrial progress [1,2,3]. The primary source of pollution is industrial, agricultural, and communal wastewater, which may adversely affect the environment and human health when inadequately treated before being discharged into the recipient [4]. Textile, graphic, food, pharmaceutical, photographic, and cosmetic industries use dyes in the production process, producing many colored wastewaters characterized by high oxygen consumption [5,6,7,8]. Even in low concentrations in the water, the presence of dyes harms aquatic organisms’ life due to their toxic, mutagenic, and carcinogenic properties [7,8,9]. In long-term contact with humans, dyes can cause severe damage to the kidneys, liver, and central nervous system. In addition, dyes may cause respiratory problems or the appearance of allergic reactions and dermatitis [7,10,11,12,13].

Methylene blue (MB) belongs to the group of xanthene dyes, and rhodamine B (RB) to the group of thiazine dyes [14]. They are used for dyeing paper and in the textile industry for dyeing cotton, leather, and wool. They have been widely used in medicine and the pharmaceutical and cosmetic industries [15,16,17]. Due to the positive charge in the structures, these dyes belong to cationic dyes [18], with the fact that RB in the aqueous environment at pH < 3 is in the protonated (cationic) form, while at pH > 3, it is in the zwitterion form due to the deprotonation of the carboxyl group [19,20,21,22]. The accumulation of cationic dyes in the body may cause respiratory problems, rapid heartbeat, nausea, vomiting, abdominal pain, and skin irritation [18,23]. They are hardly biodegradable and persist in the environment in very high concentrations [17,24]. It is, therefore, necessary to remove or reduce them to a minimum in wastewater before discharge into waterways.

Due to all the aforementioned, numerous physical, chemical, and biological methods have been developed to remove dyes from the water environment [3,25,26,27,28]. Some of them are: adsorption [4,29], membrane filtration [30], electrocoagulation [31], photochemical decomposition [32], electrochemical oxidation [33], as well as advanced oxidation processes [34,35]. In addition to many techniques used in the treatment of colored wastewater, adsorption still takes an important place due to its simplicity, practicality, economy, and high efficiency.

The adsorption efficiency mainly depends on the affinity of the adsorbate to the adsorbent and the specific surface of the adsorbent [8,36,37,38,39,40]. Activated carbon is one of the most frequently used adsorbents that dates back to ancient times. Carbon materials are still used today to purify polluted waters thanks to the surface’s high porosity and chemical reactivity [7,41]. Recently, advanced oxidation processes have been increasingly used for wastewater treatment. Within this approach, highly reactive species, such as hydroxyl radicals (^•^OH), are generated and may indiscriminately react with many organic compounds by electrophilic addition to the double bond or electron-transfer reaction. The formed intermediates can further react with dissolved molecular oxygen and increase degradation efficiency to less toxic or non-toxic compounds [42,43,44,45,46,47]. Heterogeneous photocatalytic degradation, which can completely mineralize organic compounds to CO_2_, H_2_O, and inorganic ions, has attracted particular attention [43,48].

Thanks to their high adsorption capacity, thermal and mechanical stability, and unique and precisely defined crystal structure, zeolites have been widely applied in adsorption, ion exchange, and catalysis. Recently, zeolites have been used in photodegradation processes, often as carriers of photocatalysts [49,50,51]. The nanoporous aluminosilicate zeolite lattice, which consists of tetrahedral units TO_4_ (T = Al, Si) interconnected by oxygen atoms, is negatively charged, and cations and water molecules are placed in the channels and cavities to achieve electroneutrality [7,52,53].

The properties of zeolites, such as adsorption, ion exchange, etc., are determined by the internal and external surface and by the SiO_2_/Al_2_O_3_ ratio [52]. The surfaces of high-silicate zeolites have a pronounced organophilic-hydrophobic character, unlike low-silicate and medium-silicate zeolites that are suitable adsorbents for water, ammonia, and other polar molecules and are also often used for the removal of heavy metals [51,54]. Synthetic zeolites that belong to the group of high-silicate zeolites are BETA (from the BEA group), ZSM-5 (from the MFI group), and Y (from the FAU group), which has a significantly lower SiO_2_/Al_2_O_3_ ratio.

The three-dimensional structure of the BETA zeolite is characterized by two mutually vertically flat channels (0.76 × 0.64 nm) and a sinusoidal channel with a diameter of 0.55 × 0.55 nm. In contrast, the porous structure of ZSM-5 zeolite is characterized by the flat channels of an elliptical cross-section (0.51 × 0.55 nm), which are interspersed with a sinusoidal channel of a circular cross-section (0.54 nm) [55,56]. Zeolite Y (from the FAU group) has high thermal stability and is characterized by 0.74 nm-diameter pores and a 1.2 nm central cavity [55].

This study observed the removal efficiency of MB and RB dyes from the aqueous environment by adsorption on NH_4_BETA, NH_4_ZSM-5, and NaY zeolite and by photodegradation under the influence of UV radiation in the presence of these zeolites. After photodegradation, the degree of mineralization was analyzed as chemical oxygen demand (COD) to gain a better insight into the photocatalytic process efficiency. Quantum-mechanical calculations within the density functional theory (DFT) approach gave us important insights into the reactivity of MB and RB, which helped us to interpret the obtained results, and explain the difference in degradation efficiency of these two dyes.

## 2. Results and Discussion

### 2.1. Results of Adsorption Observation

The Freundlich isotherm and specific adsorption parameters describe the adsorption suspension of RB-zeolites and MB-zeolites. To establish the most appropriate adsorption equilibrium correlation and the accuracy in parameter prediction of non-linear isotherm models were compared and discussed (Figure 1, Figure 2, Figure 3, Figure 4, Figure 5 and Figure 6).

The Freundlich adsorption isotherm is mainly used for expressing the adsorption of non-idealized systems where the adsorbent surface is energetically heterogeneous, indicating the formation of multilayers. During adsorption from the solution, the expression applies:$$q_{e}{=K}_{F}\cdot c^{1/n}$$
where $q_{e}$ is the equilibrium amount of adsorbed substance per unit mass of adsorbent (x/m), *c* is the equilibrium concentration, and *K_F_* and *n* are specific empirical constants [14,50,57]. The experimental results of adsorption observation of RB from the aqueous environment on NH_4_BETA, NH_4_ZSM-5, and NaY zeolite are presented in Figure 1, Figure 2 and Figure 3 and 7 (where *x*/*m* is the amount of adsorbed substance in equilibrium, *γ_e_* is the adsorbate concentration at equilibrium, and *γ_o_* is the initial concentration of adsorbate). Adsorption parameters (*k, n,* ∆*_ads_H_m_* presented in Appendix A) were determined from the graphs of functional dependence $\ln x/m\;\mathrm{vs}.\;\;\ln\gamma_{e}$ (Appendix A).

Based on the obtained results regarding the adsorption of RB (Figure 1, Figure 2 and Figure 3) and MB (Figure 4, Figure 5 and Figure 6) on NH_4_BETA, NH_4_ZSM-5, and NaY zeolites, one may determine that the adsorption isotherms pass through several plateaus (type of Freundlich adsorption isotherm), and according to Giles belong to S4 group, indicating multi-layered physical adsorption. The only exception occurs for the RB-NH_4_ZSM-5 suspension, at 293 K, where one plateau is registered, which suggests that the adsorption is a monolayer and that, in addition to physical adsorption, chemisorption also occurs.

Physical adsorption results from weak intermolecular interactions, while chemisorption is based on the exchange of valence electrons between the adsorbent and adsorbate. One may not always set the boundary between physisorption and chemisorption. Observing RB and MB’s adsorption on zeolites can provide interesting information about possible interactions with the zeolite surface. The results show that NaY zeolite was the worst adsorbent for RB and MB dye. Obtained results can be explained by the fact that the acidity of the surface was decisive for adsorption, i.e., the presence of Lewis and Brønsted active centers located on the outer and inner zeolite surfaces.

The higher ratio of SiO_2_/Al_2_O_3_ in NH_4_BETA and NH_4_ZSM-5 zeolites indicates the presence of active sites of the acidic character of different strengths, which favor a better interaction with these dyes. The surface of high-silicate zeolites is more homogeneous with an organophilic nature, in contrast to the zeolite surface with a lower content of SiO_2_/Al_2_O_3_, which is more selective for water and other polar molecules [51,54,58].

The greater degree of removal of RB (Figure 7) and MB (Figure 8) dyes by adsorption on NH_4_BETA zeolite compared to NH_4_ZSM-5 zeolite can be explained by the larger specific surface area of NH_4_BETA zeolite, and this is supported by the value of the constant *k* (Appendix A), which is the highest for the suspension MB-NH_4_BETA (*k* = 54.87) and RB-NH_4_BETA (*k* = 15.35); this shows a very high affinity between this adsorbent and adsorbates. Based on the value of the constant *n* (Appendix A), it can be concluded that there is a relatively strong interaction between the active centers of adsorbents and adsorbates and that it is the strongest at the temperature of 293 K for the RB-NH_4_ZSM-5 suspension.

With the increase in temperature, the adsorption capacity of NH_4_BETA and NH_4_ZSM-5 zeolites for RB dye also increases, as indicated by the total number of adsorbed molecules (by 8% on NH_4_ZSM-5; for I plateau by 63%, and for II plateau by 45% on NH_4_BETA). This could mean that the dye molecules, in addition to weak van der Waals interactions, partially bonded to the adsorbent surface even by chemical bonding. The value of the adsorption heat also confirms this for NH_4_ZSM-5 zeolite, which, although it remained within the limits of physisorption, increased with the temperature increase. RB molecules were probably attached to the active centers of the zeolite by hydrogen bonds via hydrogen atoms from hydroxide groups located on the surface of the zeolite and electronegative nitrogen or oxygen atoms in the dye structure.

The calculated value Q (Appendix A) indicates that most molecules of RB dye were adsorbed to the surface of the NH_4_ZSM-5 zeolite and that the number of adsorbed molecules increases with the increase in temperature (from 53 to 58%). In endothermic reactions, with an increasing temperature, the number of adsorbed species increases because their mobility increases. The total number of adsorbed molecules of RB dye on the NH_4_BETA zeolite at 293 K amounted to 29%, which indicates that the dye molecules on the NH_4_BETA zeolite became bonded to the active centers on the outer surface and that the large molecules of RB dye could not reach the inner surface of the zeolite, which is composed of pores and cavities.

With the increase in reaction temperature, the quantity of adsorbed RB molecules on NaY zeolite decreases, which indicates physical adsorption, and the value of adsorption heat also confirms this. Based on the data presented in Appendix A, it can be noticed that the adsorption capacity of NH_4_BETA for MB dye increases with an increasing temperature (I plateau by 23%). In contrast, the adsorption capacity of other zeolites decreases with the increase in temperature, which can also be seen based on the total number of adsorbed MB molecules on the zeolite surface. The highest value of Q is for the MB-NH_4_ZSM-5 suspension at 283 K, but this value decreased with the temperature increase. Additionally, the number of total adsorbed MB molecules on the surface of the NaY zeolite decreased by 37% with the increased adsorption temperature, indicating that the adsorbate molecules became tied to the active centers by van der Waals forces or dipole bonds. This is also confirmed by the calculated values of adsorption heat (∆_ads_H_m_), which decreased with the increase in adsorption temperature.

When we compare RB and MB dyes, the higher degree of RB removal in the presence of NH_4_ZSM-5 zeolite can be explained by the greater possibilities of binding to the outer zeolite surface because at pH > 3, RB dye behaves both as a weak base and as a weak acid. The high degree of MB dye removal in the presence of NH_4_BETA and NaY zeolites can be explained by the smaller dimensions of the MB molecules, which, in addition to binding to the outer surface, are most probably also bound to the active centers of the inner zeolite surface [17,49,59].

### 2.2. Photodegradation and Mineralization

When considering the photocatalytic activity of zeolites, many studies focus on zeolites containing framework heteroatoms or a modification of zeolites with oxides or metal ions precisely because of the improvement in photocatalytic capabilities [60]. This is evident from research in which zeolites were used as supports for TiO_2_ photocatalysts [61,62] or zeolite-based composites were doped with metal ions [62], whereby a higher efficiency of dye degradation was achieved than in the presence of zeolites and TiO_2_ individually. However, one must not ignore the fact that zeolites are crystalline aluminosilicates built from tetrahedra of silicon and aluminum interconnected through oxygen atoms, and it is generally known that porous SiO_2_ exhibits photocatalytic activity via siloxane bridges, which generate siloxy radicals under the influence of UV radiation (below 390 nm) [63,64] accordingly, in this manuscript, the photocatalytic activity of the original samples of synthetic zeolites in the removal of RB and MB dyes was tested.

First, the photolysis of RB and MB was examined under UV radiation to see the contribution of adding zeolites to these systems (Appendix A). Slightly higher efficiency of direct photolysis was observed at a higher temperature in the case of both dyes. However, direct photolysis is significantly less efficient in the case of both dyes compared to the system with zeolites (Figure 9 and Figure 10). The results of photodegradation of RB and MB dyes in the presence of NH_4_BETA, NH_4_ZSM-5, and NaY zeolites, under the influence of UV radiation for 180 min, are given in Figure 9 and Figure 10.

One of the critical roles in the process of photocatalytic degradation is played by adsorption, which mainly depends on the affinity of the catalyst for the substrate, the specific surface area of the catalyst, and the nature of the solvent. After adsorption, the photodegradation of adsorbed molecules commences under the influence of radiation. As already mentioned, zeolites with high SiO_2_/Al_2_O_3_ content, such as NH_4_BETA and NH_4_ZSM-5, show a high affinity for removing dyes.

The obtained results of photodegradation of MB and RB dyes under UV radiation (Figure 9 and Figure 10) indicate that the highest efficiency of photodegradation at 283 K was achieved in the presence of NH_4_ZSM-5 zeolite, then NH_4_BETA, and finally NaY zeolite. It can be seen in Appendix A that NH_4_ZSM-5 zeolite has the highest ratio of SiO_2_ and NaY has the lowest.

The presence of acid-base centers in the zeolite structure, which have electron donor and electron acceptor properties, prevents the possible recombination of electrons and holes, leading to a higher photodegradation efficiency. The results of the photodegradation of MB dye in the presence of zeolite at 293 K follow the same trend as at 283 K. The increase in temperature did not significantly affect the degree of degradation of MB dye in the presence of NH_4_BETA zeolite. In contrast, in the presence of NH_4_ZSM-5 and NaY zeolite, there is an insignificant decrease in the degree of degradation, which is probably the effect of the physisorption of the dye molecules on the catalyst surface. In the case of RB dye, the highest efficiency of photodegradation at 293 K was achieved in the presence of NH_4_BETA zeolite (83.3%). It can be observed that the degree of removal of RB is increased by the increase in temperature reaction, which is most likely the effect of the increased frequency of molecular collisions in the solution.

The photodegradation of MB and RB molecules in the presence of zeolite under UV radiation includes the generation of electron and hole pairs. Electrons in the conduction zone are unstable and pass to adsorbed oxygen molecules, where superoxide radicals are formed, while holes in the valence zone can be captured by the molecules of dye or water, creating ^•^OH. The adsorbed dye molecules can be degraded by radical species and mineralized into less toxic products.

Based on the structures of RB and MB dyes, it can be expected that depending on the degree of mineralization, in addition to CO_2_ and H_2_O, inorganic ions ${\mathrm{NH}}_{4}^{+}$, ${\mathrm{NO}}_{3}^{+}$, and ${\mathrm{SO}}_{4}^{2-}$ are also formed. According to the results shown in Figure 11 and Figure 12, it can be concluded that the highest degree of mineralization of MB dye (at 283 K) and RB dye (at 283 K and 293 K) was achieved in the presence of NH_4_ZSM-5, then NH_4_BETA, and the lowest degree of mineralization was achieved in the presence of the NaY zeolite. The obtained results of mineralization agree with the results of photodegradation. The mineralization degree is lower than the degree of photodegradation, which indicates the presence of various intermediates, whose mineralization is often slower than the degradation of the initial compound. In the case of MB dye, the degree of mineralization at 293 K is the highest in NH_4_ZSM-5 zeolite and the lowest in NH_4_BETA zeolite.

The presence of multiple intermediates is also indicated by the change in the pH value of the solution during the photocatalytic process. As seen in Table 1, during the photodegradation of RB dye in the presence of all zeolites, the pH value decreases (due to the formation of acidic intermediates), except in the presence of NaY zeolite at 293 K, where the pH value increases. During the decomposition of MB dye, the pH value decreases in the presence of NH_4_ZSM-5 zeolite, as well as NH_4_BETA (283 K), while in the presence of the NaY zeolite, the pH value increases due to the formation of base intermediates.

During the decomposition of RB dye, radical species probably first attack nitrogen atoms (N-deethylation process), whereby various intermediates with an aromatic ring are formed. After that, reactive radicals attack carbon atoms, resulting in the cleavage of chromophores and oxidation of intermediates to carboxylic acids, preventing the increase of pH value, as per the results presented in Table 1. Decomposition of a -C-S=S- functional group starts with the electrophilic attack of radical species on the -C-S=S- functional group in the structure of the MB dye molecule, where sulfate ions are formed through sulfoxide, sulfone, and sulfonic acid. In addition, the attack of radicals on nitrogen atoms can lead to the formation of substituted anilines, phenols, aldehydes, or carboxylic acids, which can further be mineralized to CO_2_ and H_2_O. The degree of mineralization of RB dye is lower than that of MB dye (except for RB-NH_4_BETA at 293 K), which is probably a consequence of its forming a more significant number of intermediates. The appearance of a more significant number of intermediates in the case of RB dye is expected, given that the RB molecule is larger than the MB dye.

### 2.3. Computational Analysis

To better understand the experimental results, the influence of the structure of the tested dyes was analyzed. MB and RB are dyes that consist of cation and Cl anion. The starting geometries of MB and RB have been taken from the ChemSpider library and subjected to geometrical optimizations. Since there is a noncovalent interaction between the cation and anion in these cases, we had to apply the dispersion-corrected variant of the B3LYP functional. The geometrically optimized structures of MB and RB are presented in Figure 13, with the indicated distances between the cation and Cl anion.

In both cases, the Cl anion was placed near the cations’ positively charged atoms. This means that in the case of the MB, the Cl anion was placed above the sulfur atom, while in the case of the RB, the Cl anion was placed above the nitrogen atom. In the case of the MB, the Cl anion moved significantly to the plane corresponding to the central ring and shifted to interact with the nearby hydrogen atom noncovalently. In the case of the RB, the Cl anion practically remained in the vicinity of the positively charged nitrogen, which disagrees with the finding reported by Delgado and Selsby [65]. Namely, in their computational study, the Cl anion shifted towards the central part of the RB. However, the disagreement between these results is expected since they applied the Hartree–Fock method for geometrical optimizations, which is the level of theory that neglects the electron correlation. There is a significant difference in terms of the shortest distance between the Cl anion and the cationic fragment since the Cl anion is much closer to the cationic fragment in the case of the MB dye.

Computational analysis in this work encompassed calculations of the MEP and ALIE quantities, two well-known quantum molecular descriptors describing the local reactivity of the molecules. The MEP descriptor is one of the most frequently calculated quantities to identify molecules’ sites with electron abundance or deficiency. Another popular descriptor for addressing the local reactivity of molecules is the ALIE quantity. MEP is a substantial quantity that reveals which molecular sites are prone to interact with other molecules based on electrostatic interactions. One molecule’s electron-abundant site would react with the electron-deficient site of the other molecule. An equally important but somewhat less frequently applied descriptor is ALIE, which reveals the molecular sites where the lowest amount of energy is required to remove an electron. In other words, this descriptor indicates the molecular sites sensitive to electrophilic attacks. Essentially, both MEP and ALIE descriptors were introduced into practice by Professor Politzer and his coworkers [66,67,68,69,70]. The most practical way to analyze the values of MEP and ALIE descriptors is by their mapping to the electron density surface, which was performed in this work. MEP and ALIE surfaces of the MB and RB dyes are presented in Figure 14.

The analysis of the minimal and maximal values of the MEP descriptor provides a further understanding of the results related to dye removal via adsorption. Namely, the lowest MEP values for both MB and RB dyes are practically the same and are equal to −75.73 kcal/mol and −72.91 kcal/mol, respectively. The similar lowest MEP values are expected since the Cl anion bears the negative charge in both cases. However, there is a huge difference in the maximal MEP values in favor of the RB dye. Namely, the MB dye’s highest MEP value is 34.35 kcal/mol, while the RB dye’s highest MEP value is 56.16 kcal/mol. This discrepancy in the highest MEP values between the MB and RB dyes shows that the RB dye is far more prone to interact with other structures based on electrostatic interactions. Indeed, the adsorption study in this work indicates that the RB dye is removed to a greater extent than the MB, which agrees with the computational results. Conversely, the minimal and maximal ALIE values of MB and RB dyes are very similar, so they are expected to have comparable sensitivity toward electrophilic attacks.

## 3. Materials and Methods

### 3.1. Chemicals and Solutions

The standard dye solutions (0.5–5 mg/dm^3^) of MB and RB (products of the Merck company, Germany) were prepared by diluting the basic solutions. Thehe absorption maxima were then determined, and calibration curves were constructed (Appendix A). Synthetic zeolites were used during the work (products of Zeolyst International company, Kansas City, KS, USA): NH_4_BETA (from the BEA group, mark CP 814E), NH_4_ZSM-5 (from the MFI group, mark CBV 3024E), and NaY (from the FAU group, mark CBV 100).

### 3.2. Structural Analysis of Zeolites

All zeolites’ specific surfaces (Sp) were determined by the BET method on the Flowsorb II–2300 instrument (Appendix A).

Additionally, the FT-IR spectra of all zeolites were recorded on the Thermo Scientific Nicolet iS10 FT-IR Spectrometer with a resolution of 4 cm^−1^ and 32 scans to identify the surface acidity of the zeolite (Appendix A).

### 3.3. Adsorption Experiments

Adsorption experiments of the dyes MB and RB from the aqueous environment on NH_4_BETA, NH_4_ZSM-5, and the NaY zeolite were performed at 283 K and 293 K for 180 min. iThe reaction suspension was always thermostated (Thermostat: WiseCircu WCR, model WCR-P22, Witeg, Wertheim, Germany). The adsorbent mass was 0.25 g (exact weight ±10^−4^ g) in contact with 50cm^3^ of adsorbate solutions of different concentrations (0.5–5 mg/dm^3^). The results are presented as curves of functional dependence x/m on the concentration at equilibrium *γ*_e.,_ from which the quantity of adsorbed dye on one adsorption monolayer was read, i.e., on the plateau. The amount of adsorbed dye (x) was calculated from the difference in concentrations before and after adsorption, while x/m represents the quantity of adsorbed dye per adsorbent mass unit. Other adsorption parameters, the number of adsorbed molecules on individual plateaus (*N*) and their total surface (*S*), as well as the parameter *Q*, which represents the ratio of the surface of adsorbed molecules and the specific surface of the zeolite, were calculated. The parameters *k* and *n*, as well as the adsorption heat ($∆H_{\mathrm{ads}}$), were determined by calculations and graphic presentation (from the diagram of dependence $\ln x/m\;\mathrm{vs}.\;\;\ln\gamma_{e}$).

### 3.4. Photodegradation

Photodegradation of the dyes MB (1 mg/dm^3^) and RB (1 mg/dm^3^) under UV radiation was performed in a photochemical cell made of double-walled Pyrex glass in the presence of NH_4_BETA, NH_4_ZSM-5, and the NaY zeolite. Into the cell, 20 cm^3^ of the aqueous solution of dye and 0.1 g of zeolite were weighed (exact weight ±10^−4^ g), followed by sonification in an ultrasonic bath for 15 min, to achieve a uniform size of the catalyst particles. The photochemical cell was placed on a magnetic stirrer, and the suspension was continuously mixed during the irradiation with a constant oxygen flow. Photodegradation was performed at 283 K and 293 K for 180 min and a high-pressure mercury lamp with a suitable concave mirror was used as a source of UV radiation (Philips, HPL-N, 125 W, with emission bands in the area of UVA radiation at 304, 314, 335 and 366 nm, with an emission maximum at 366 nm).

### 3.5. Analytical Procedures

Dye concentrations before and after adsorption were determined spectrophotometrically on the Perkin Elmer UV/VIS Spectrometer Lambda 25 instrument.

The concentration of dyes in specific time intervals after photodegradation was determined on the double-beam T80 + UV-vis Spectrometer (UK), at a fixed slit width (2 nm), using a 1 cm quartz cell and computer-loaded UV Win 5 data software.

The pH value of standard dye solutions was determined using a glass electrode (AmpHel pH electrode, Hanna Instruments, Cluj Napoca, Romania) connected to the pH meter (Bench pH meters, Hanna Instruments, Cluj Napoca, Romania). For monitoring the pH during the degradation, a combined glass electrode (pH-Electrode SenTix 20, Xylem Analytics Germany Sales GmbH & Co. KG, WTW, Weilheim, Germany) connected to the pH meter was used (pH/Cond 340i, Xylem Analytics Germany Sales GmbH & Co. KG, WTW, Weilheim, Germany).

The COD was determined according to Standard Method 410.4 declared by EPA, United States Environmental Protection Agency. The calibration curve was obtained using HOOCC_6_H_4_COOK as a standard solution, and *R^2^* was 0.995 (Appendix A). Aliquots of 2.5 cm^3^ were taken from the reaction mixture after 90 min of photodegradation experiments. The COD concentration was determined spectrophotometrically by measuring the absorbance of the formed Cr^3+^ at a fixed slit width (2 nm) using a quartz cell (1 cm optical length) and computer-loaded UV Win 5 data software. Absorbance was recorded on a double-beam T80 + UV-vis Spectrometer. The evolution of absorbance of the digested solution was recorded at 600 nm. The samples were digested at 150 °C for 2 h.

### 3.6. Computational Methods

Molecular DFT calculations on the RB and MB were performed by applying the dispersion corrected version of a B3LYP functional [71], namely the B3LYP-D3 variant [71,72], in combination with the 6–31G(d,p) basis set [73,74]. At the mentioned level of theory, the structures of RB and MB were first geometrically optimized to reach the ground states. Frequency calculations were further performed to ensure that the actual equilibrium states were identified, confirmed by the inexistence of the imaginary frequencies. After obtaining the ground states of the MB and RB, molecular electrostatic potential (MEP) and average local ionization energy (ALIE) were calculated to analyze the local reactivity properties of dyes, using the M06-2X functional [75] and the same basis set.

Molecular DFT calculations related to geometrical optimizations and vibrational analysis were performed with an ORCA 5.0.3. package [76,77,78,79]. Input files for ORCA were made with the atomistica.online web application [80]. Calculations of the MEP and ALIE were performed with the Jaguar program [81,82,83], as implemented in the Schrödinger Materials Science Suite 2022-2. SAPT0 calculations were performed with the PSI4 modeling program [84,85,86].

## 4. Conclusions

Zeolites have a significant place among the most commonly used adsorbents thanks to their specific crystal structure consisting of pores, channels, and cavities of different dimensions, which results in a large internal surface area available for the removal of various organic pollutants, including dyes. The properties of zeolite are determined, among other things, by active centers of different acidity (Lewis and Brønsted type) located on the outer and inner surfaces of the zeolite. Recently, zeolites have increasingly been used as catalysts in photodegradation processes. This study observed the adsorption and photodegradation of the selected organic dyes, MB and RB, in the presence of zeolite material from BEA, MFI, and FAU groups. The characterization of all adsorption suspension s is given through the Freundlich adsorption isotherm, and the corresponding parameters were determined. The results showed that adsorption is still one of the leading methods in wastewater treatment because a high degree of dye removal was achieved under these experimental conditions. NH_4_BETA and NH_4_ZSM-5 zeolites are very effective adsorbents for these dyes due to acid centers of different strengths responsible for the adsorption, while the NaY zeolite proved to be the least effective. The higher degree of adsorption on the NH_4_BETA zeolite compared to NH_4_ZSM-5 can be attributed to a higher specific surface area (about 40% higher). Adsorption of dyes on all zeolites occurs according to the principle of multi-layer physical adsorption, except for the RB dye on NH_4_ZSM-5, where chemisorption partially occurs. In addition, the high efficiency of removing these organic pollutants was achieved by heterogeneous photocatalysis, where the highest degree of photodegradation of dyes was performed in the presence of a NH_4_ZSM-5 zeolite (highest SiO_2_/Al_2_O_3_) primarily and secondarily in the presence of a NH_4_BETA zeolite, probably due to the formation of siloxy radicals. Siloxy radicals with hydroxyl and superoxide radicals contributed to a high degree of dye degradation. Compared to the photodegradation efficiency of dyes, the lower degree of mineralization could be explained by the formation of various decomposition intermediates, which requires a longer irradiation time.

## Figures and Tables

**Figure 1 molecules-27-06582-f001:**
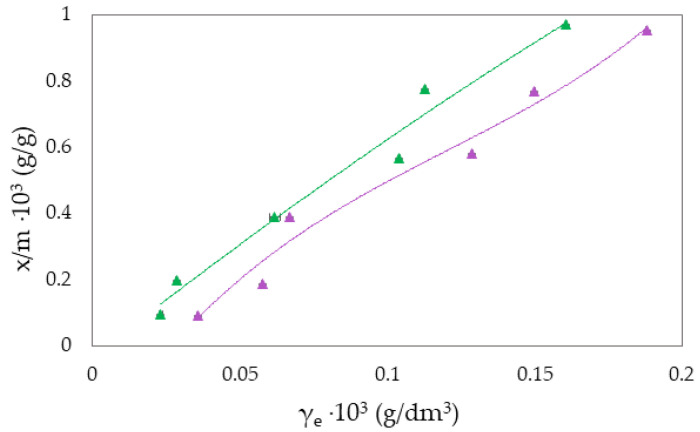
Freundlich adsorption isotherm curves for the suspension RB-NH_4_BETA at ▲283 K and ▲293 K.

**Figure 2 molecules-27-06582-f002:**
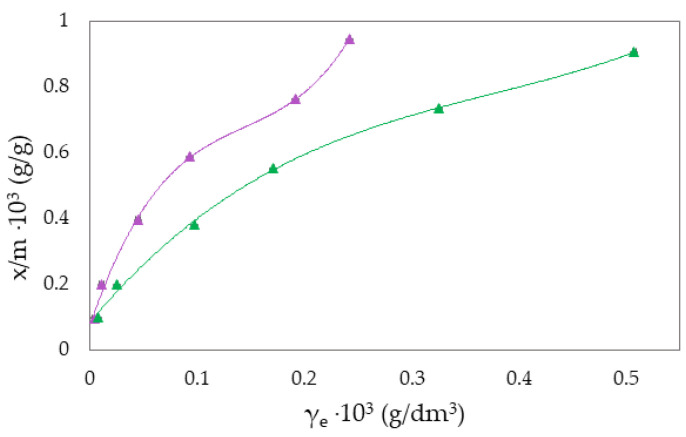
Freundlich adsorption isotherm curves for the suspension RB-NH_4_ZSM-5 at ▲283 K and ▲293 K.

**Figure 3 molecules-27-06582-f003:**
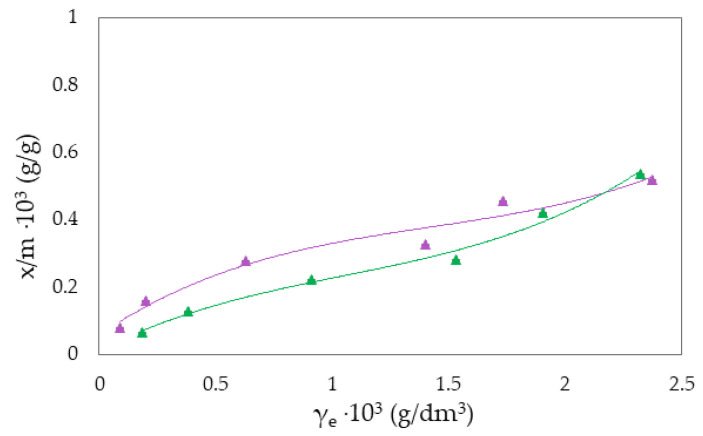
Freundlich adsorption isotherm curves for the suspension RB-NaY at ▲283 K and ▲293 K.

**Figure 4 molecules-27-06582-f004:**
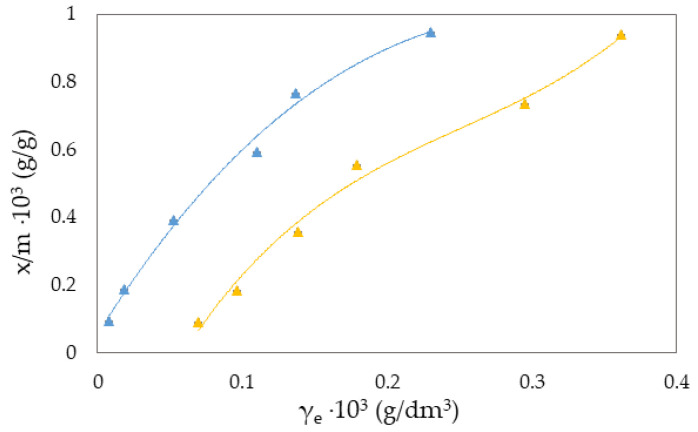
Freundlich adsorption isotherm curves for the suspension MB-NH_4_BETA at ▲283 K and ▲293 K.

**Figure 5 molecules-27-06582-f005:**
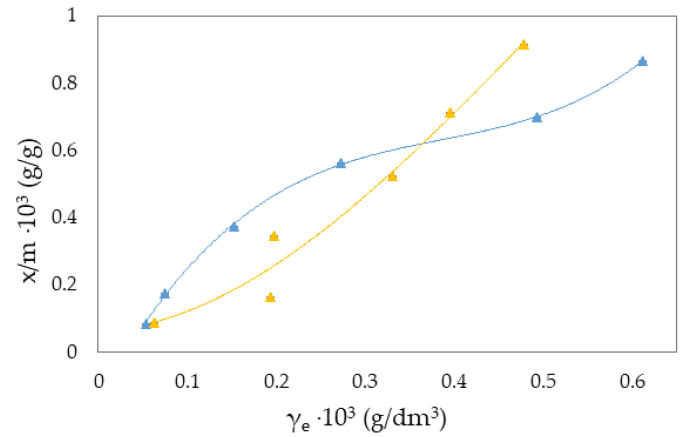
Freundlich adsorption isotherm curves for the suspension MB-NH_4_ZSM-5 at ▲283 K and ▲293 K.

**Figure 6 molecules-27-06582-f006:**
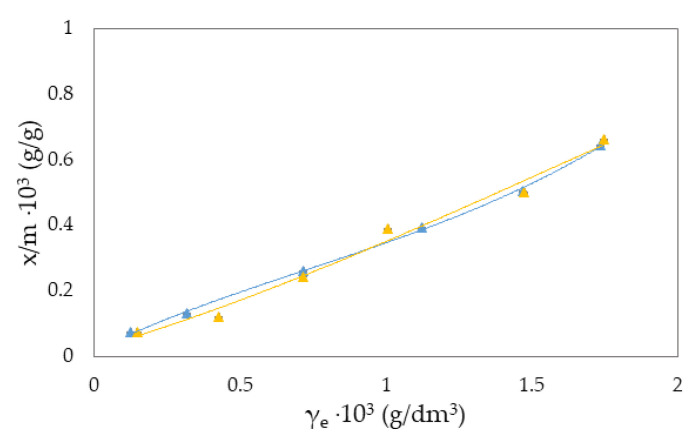
Freundlich adsorption isotherm curves for the suspension MB-NaY at ▲283 K and ▲293 K.

**Figure 7 molecules-27-06582-f007:**
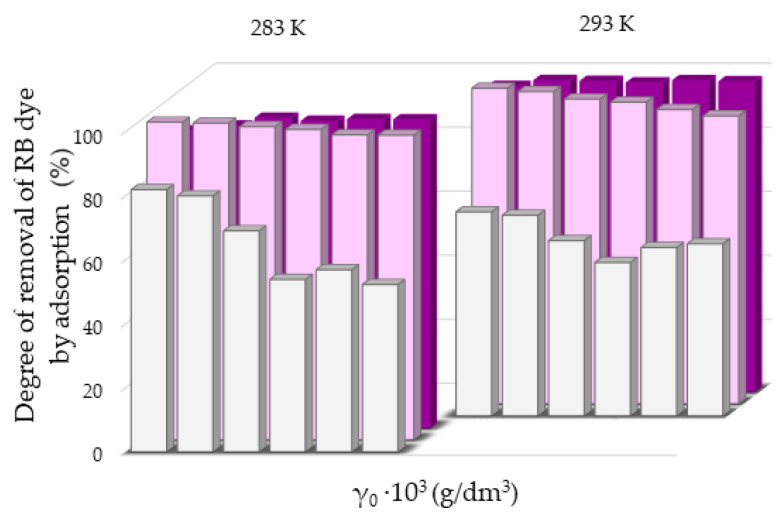
Removal degree of RB dye by adsorption on ■NaY, ■NH_4_ZSM-5 and ■NH_4_BETA zeolites at two temperatures, calculated after 180 min.

**Figure 8 molecules-27-06582-f008:**
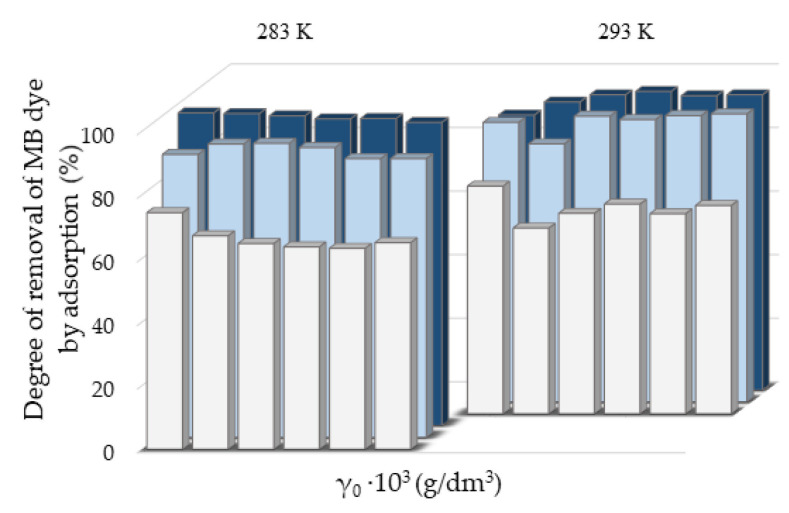
Removal degree of MB dye by adsorption on ■NaY, ■NH_4_ZSM-5, and ■NH_4_BETA zeolites at two temperatures, calculated after 180 min.

**Figure 9 molecules-27-06582-f009:**
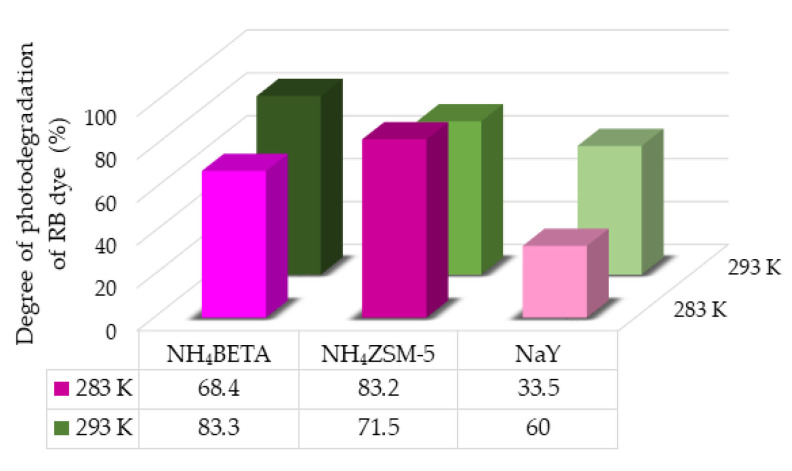
Degree of photodegradation of RB dye in the presence of selected zeolites under the UV radiation at two temperatures after 180 min.

**Figure 10 molecules-27-06582-f010:**
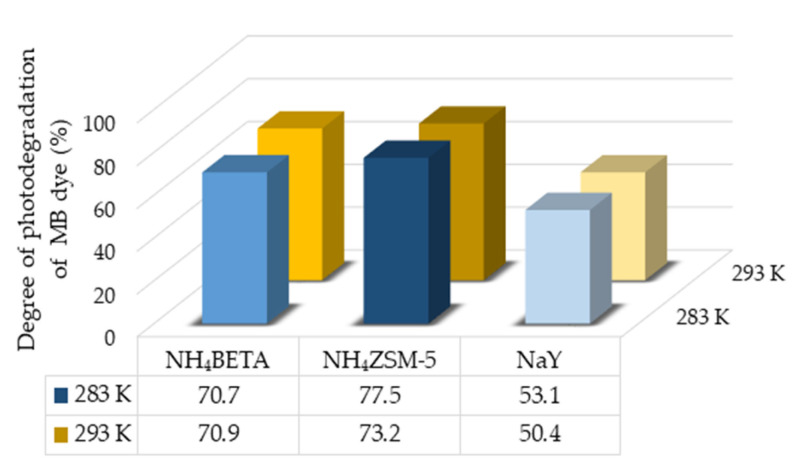
Degree of photodegradation of MB dye in the presence of selected zeolites under the UV radiation at two temperatures after 180 min.

**Figure 11 molecules-27-06582-f011:**
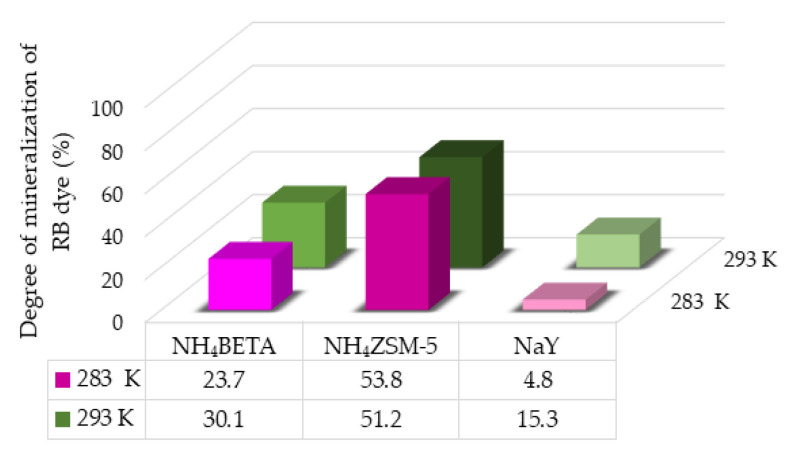
Mineralization degree of RB dye in the presence of selected zeolites after 180 min under UV radiation at two temperatures.

**Figure 12 molecules-27-06582-f012:**
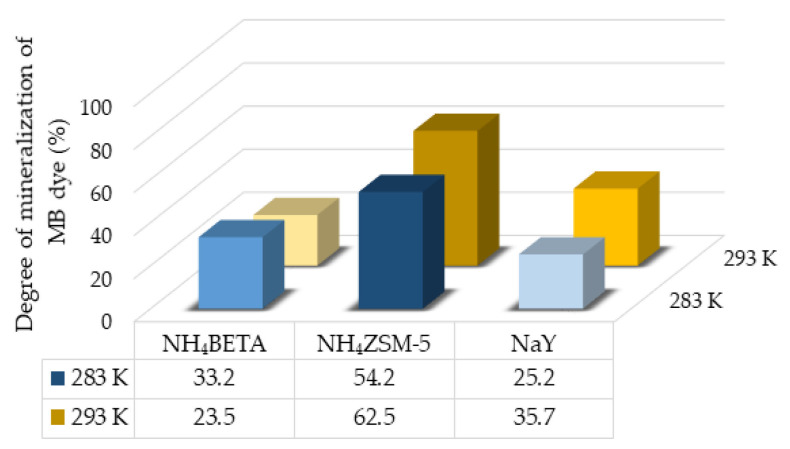
Mineralization degree of MB dye in the presence of selected zeolites after 180 min under UV radiation at two temperatures.

**Figure 13 molecules-27-06582-f013:**
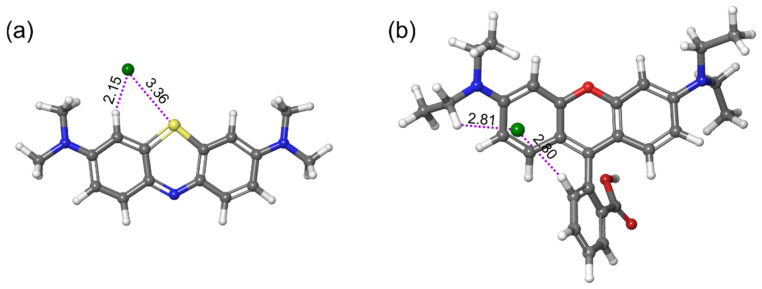
Optimized structures of (**a**) MB and (**b**) RB with distances (in Å) between anion and cation.

**Figure 14 molecules-27-06582-f014:**
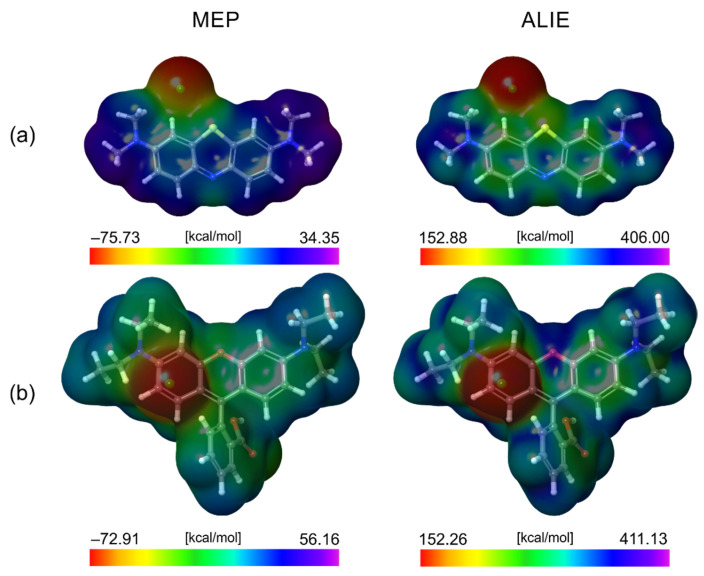
MEP and ALIE surfaces of (**a**) MB and (**b**) RB dyes.

**Table 1 molecules-27-06582-t001:** The change in pH value of the solution during the photodegradation dyes (1.0 mg/dm^3^) in the presence of 5.0 mg/cm^3^ zeolite under UV radiation.

Time (min)	RB	pH at 283 K	pH at 293 K	MB	pH at 283 K	pH at 293 K
0	NH_4_BETA	7.3	7.2	NH_4_BETA	7.0	6.9
60		7.0	6.9		6.9	6.9
180		6.8	6.9		6.9	7.0
0	NH_4_ZSM-5	7.1	7.1	NH_4_ZSM-5	6.3	6.6
60		6.1	6.9		6.1	5.8
180		6.2	6.9		6.2	5.8
0	NaY	8.7	8.5	NaY	8.5	8.4
60		8.4	8.2		8.7	8.8
180		8.3	8.6		8.7	9.0

## Data Availability

Not applicable.

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
