# Peer review of "Zeolites as Adsorbents and Photocatalysts for Removal of Dyes from the Aqueous Environment"

_molecules, 2022, doi:10.3390/molecules27196582_

Round 1
Reviewer 1 Report
This work deals with the use of commercial zeolites to remove dyes from water by adsorption and photocatalysis. The scope of this study is noteworthy, highlighting the computational analysis that is presented. I consider that the work is interesting for the readers of Molecules, but there are some issues in the manuscript that need to be clarified and completed, which requires a major revision.
The main issues that the authors should note:
First, I don´t understand the use of the word “composite”, present in the title and abstract. The definition of “composite is: “material made from two or more different materials that, when combined, are stronger than those individual materials by themselves”, so I don’t understand why the authors use this word since materials used in the study are solely commercial zeolites where NH4+ is the compensation cation of the framework in the case of BEA and ZSM-5 and Na+ in the case of Y (this later information is missing in the case of Y and should also be included in the designation of the sample). I recommend the removal of the word “composite” since it may mislead the reader.
Introduction Section: The materials that are mostly used as adsorbents are carbons or activated carbons, not zeolites, that are limited by their intrinsic microporosity that cause diffusional limitations to voluminous molecules in liquid phase, which clearly happens in this study. The authors should at least mention the use of carbon materials as adsorbents.
Results and discussion
Results of adsorption observation: Although the Freundlish isotherm model is widely known, the equation should appear in the manuscript as well as some contextualization. Why was this isotherm model selected? Other isotherm models were tested, for example Langmuir isotherm model? Please, clarify.
Concerning Fig 1-6 I must call the attention to the plots. First, the number of experimental points is very reduced to establish the correct configuration of the isotherm, as the authors present (lines 117-122). On the other hand, some statistical parameters should be displayed to show how good the experimental points fit the isotherm profile.
The authors should relate the adsorption and photocatlytic behavior of each zeolite structure with the two dye molecules. I suggest the following topics for discussion: i) two of the zeolite structure are in NH4+ form and another (Y) in sodic form, with evident implications in the acidity properties and interaction with the dye molecules ii) two large pore zeolite or 12 MR (BEA and Y) and a medium pore size or 10 MR zeolite (ZSM-5) with clear influence on the access and diffusion of the dye molecules inside the pores, especially considering the distinct dimensions of the dye molecules.
Author Response
Journal Molecules (ISSN 1420-3049)
Manuscript ID molecules-1897082
Title: Zeolite-based composite for removal of dyes from water by adsorption and photocatalysis
Special Issue Modeling Adsorption Properties of Molecular and Nanostructured Systems for Environmental Applications
The authors thank the reviewer for their constructive comments and recommendations. We have taken the comments on board to improve and clarify the Manuscript. Please find below a detailed point-by-point response to all comments.
The reviewers’ comments are typed in black. The responses to reviewers are tagged blue. The changes in the text of the Manuscript are typed in red. The line numbers in responses to reviewers refer to the corrected text.
Response Reviewer 1:
This work deals with the use of commercial zeolites to remove dyes from water by adsorption and photocatalysis. The scope of this study is noteworthy, highlighting the computational analysis that is presented. I consider that the work is interesting for the readers of Molecules, but there are some issues in the manuscript that need to be clarified and completed, which requires a major revision.
We thank the reviewer for the suggestions and the opportunity to improve our manuscript.
First, I don´t understand the use of the word “composite”, present in the title and abstract. The definition of “composite is: “material made from two or more different materials that, when combined, are stronger than those individual materials by themselves”, so I don’t understand why the authors use this word since materials used in the study are solely commercial zeolites where NH4+ is the compensation cation of the framework in the case of BEA and ZSM-5 and Na+ in the case of Y (this later information is missing in the case of Y and should also be included in the designation of the sample). I recommend the removal of the word “composite” since it may mislead the reader.
We understand your objection. The word "composite" has been removed from the title and abstract. We have also adopted your suggestion to label the sodium form of Y zeolite with NaY.
Introduction Section: The materials that are mostly used as adsorbents are carbons or activated carbons, not zeolites, that are limited by their intrinsic microporosity that cause diffusional limitations to voluminous molecules in liquid phase, which clearly happens in this study. The authors should at least mention the use of carbon materials as adsorbents.
We considered your suggestion and mentioned using carbon materials as adsorbents in the Introduction (Lines 69-72).
Results and discussion
Results of adsorption observation: Although the Freundlish isotherm model is widely known, the equation should appear in the manuscript as well as some contextualization. Why was this isotherm model selected? Other isotherm models were tested, for example Langmuir isotherm model? Please, clarify.
We have added the Freundlich equation in the manuscript according to the suggestion (Lines 113-120). This isotherm model has been chosen because it is the most appropriate mathematical model for observing adsorption from solutions. We also examined the Langmuir isotherm model. However, the obtained values of correlation coefficients R2 were far less than those of Freundlich correlation coefficients. In practice, the surfaces of adsorbents are not energetically homogeneous, which limits the use of Langmuir adsorption isotherm. Unlike the Langmuir adsorption isotherm, the Freundlich adsorption isotherm can be applied for non-ideal adsorption on heterogeneous surfaces. Therefore, we concluded that the Freundlich adsorption isotherm model is more suitable for the adsorption process in our case.
Concerning Fig 1-6 I must call the attention to the plots. First, the number of experimental points is very reduced to establish the correct configuration of the isotherm, as the authors present (lines 117-122). On the other hand, some statistical parameters should be displayed to show how good the experimental points fit the isotherm profile.
The authors are very grateful to the reviewer for this comment. In the literature, we have seen examples with the use of the same or less number of concentrations (less number of points). Some of them are:
https://doi.org/10.3390/molecules27175567
https://doi.org/10.3390/molecules27165160
https://doi.org/10.3390/molecules27154840
And we planned our experiment accordingly. We hope that the respected reviewer will not strongly object to this way of thinking of the author.
We subsequently added graphs of the functional dependence ln x/m (ie. ln qe) of ln γe in the Supplementary materials (Figures S1-S12), from which we obtained the Freundlich correlation coefficients. From the intercepts and slopes, we calculated the adsorption parameters (k, n, ∆adsHm) given in Tables S1 and S2. Based on the above data, we believe that the obtained results and the number of experimental points are relevant to the conclusions we presented and supported by the literature.
The authors should relate the adsorption and photocatlytic behavior of each zeolite structure with the two dye molecules. I suggest the following topics for discussion: i) two of the zeolite structure are in NH4+ form and another (Y) in sodic form, with evident implications in the acidity properties and interaction with the dye molecules ii) two large pore zeolite or 12 MR (BEA and Y) and a medium pore size or 10 MR zeolite (ZSM-5) with clear influence on the access and diffusion of the dye molecules inside the pores, especially considering the distinct dimensions of the dye molecules.
We are glad you have addressed the topic. As for the influence of the zeolite surface's acidity on the adsorption and photodegradation of dyes, a higher ratio of SiO2/Al2O3 (25 for NH4BETA; 30 for NH4ZSM-5) indicates the presence of active sites with an acid character that favor the adsorption and subsequent photodegradation of these cationic dyes, in contrast to NaY (5.10), which proved to be the least suitable adsorbent. In the manuscript, we subsequently discussed the influence of different sizes of dye molecules and zeolite pores on the removal efficiency (Lines 212-218).
Reviewer 2 Report
The manuscript describes the preparation, characterization and use of a set of simply-exchanged zeolites for the removal of dye pollutant from aqueous solutions through adsorption and photo-induced catalytic degradation.
The text is clear, quite easy to read and rich in experimental details. The Authors added a section with DFT calculation to further support the adsorption results recorded in real tests.
The study may attract attention and the results obtained in terms of removal of classical test molecules such as methylene blue and rhodamine B are interesting.
The manuscript might therefore be positively considered for publication. However, some doubtful points need some ameliorations. In detail:
1) Abstract. NH4BETA, NH4ZSM-5 and Y are referred as “composites”. Typically, this term is used for organic/inorganic hybrid materials or similar. What is meant for composite, here?
2) Some comparison of the proposed systems with the previous state of the art should be carried out. In previous works, modified zeolites have proved to be able to adsorb and mineralize model dyes very efficiently under UV or also visible light, with very high removal efficiencies too. Some examples are: https://doi.org/10.1021/ie060641o; ; https://doi.org/10.3390/ijms23031730; https://doi.org/10.3390/catal7110344; https://doi.org/10.3390/ijms23031730. Can the Authors comment on this and discuss what is the advantage of this system with respect to the previous ones?
3) There is no clear definition of “degree of mineralization” and how it is computed in Section 3. In the introduction it seems that mineralization is a different parameter than the COD. A clearer explanation for this is welcome.
4) Fig. 1 and others. The shape of the solid line passing through the points of the Freundlich isotherms is quite unusual. Must the line pass through the points? Can the line simply interpolate the points? Which is the error bar for each point?
5) Line 132. The different behaviour of NH4BETA and NH4ZSM-5 is attributed to the presence of acid active sites of various strength. Are there also different site densities across the various zeolites? The number of sites can have an influence on the number of molecules accommodated within the pores.
6) Line 145. The different degree of dye removal is explained by the larger specific surface area of NH4BETA zeolite. What about the specific pore volume of these solids (micro + mesopores)? have the Authors data about this?
7) Fig. 9 and 10. Data for blank tests (in the absence of zeolite catalyst) should be added, to evaluate the contribution of self-photodegradation (if any).
8) Line 234. The Authors invoke the role by adsorbed oxygen molecules. Did the Authors try to perform the tests under inert atmosphere?
9) Table 1. Which is the uncertainty of the pH values reported? I guess the pH value at 0 min for NH4ZSM-5, even if at different temperatures, should be very similar. On the contrary, two quite different values were obtained: 7.12 and 6.36. If so, are the rest of the variations significant enough? In some cases, pH increase or diminution is only 0.1.
10) Section 2.3. It is not clear how relevant is the first part of the computational analysis. In all examples, Cl counterion is placed in a well-determined position. However, during the adsorption process in real experiments, Cl- anion is freely dissolved in the aqueous medium and does not likely enter the zeolite channels, as it would be repelled by the mainly anionic framework of the BEA, MFI or FAU, caused by Al sites. Are the reddish/brown areas in Fig. 14 relevant to explain the real behaviour in the experimental tests? In addition, the Authors mention a “sensitivity toward electrophilic attacks”. In the adsorption process, there should be a prevalent electrostatic attraction between the positively-charged dye and the negatively-charged zeolite cavities. What is the electrophilic species that is responsible for the attack? Please, explain.
After these important points are taken carefully into account, the contribution can be considered for publication.
Author Response
Journal Molecules (ISSN 1420-3049)
Manuscript ID molecules-1897082
Title: Zeolite-based composite for removal of dyes from water by adsorption and photocatalysis
Special Issue Modeling Adsorption Properties of Molecular and Nanostructured Systems for Environmental Applications
The authors thank the reviewer for their constructive comments and recommendations. We have taken the comments on board to improve and clarify the Manuscript. Please find below a detailed point-by-point response to all comments.
The reviewers’ comments are typed in black. The responses to reviewers are tagged blue. The changes in the text of the Manuscript are typed in red. The line numbers in responses to reviewers refer to the corrected text.
Response Reviewer 2:
The manuscript describes the preparation, characterization and use of a set of simply-exchanged zeolites for the removal of dye pollutant from aqueous solutions through adsorption and photo-induced catalytic degradation.
The text is clear, quite easy to read and rich in experimental details. The Authors added a section with DFT calculation to further support the adsorption results recorded in real tests.
The study may attract attention and the results obtained in terms of removal of classical test molecules such as methylene blue and rhodamine B are interesting.
The manuscript might therefore be positively considered for publication. However, some doubtful points need some ameliorations.
The authors thank the Reviewer very much for this comment and positive opinion of our results.
Abstract. NH4BETA, NH4ZSM-5 and Y are referred as “composites”. Typically, this term is used for organic/inorganic hybrid materials or similar. What is meant for composite, here?
In our research, we used original samples of synthetic zeolites from an American company Zeolyst International, which we did not modify. Accordingly, we removed the term „composite“ from the manuscript and corrected the paper's title.
Some comparison of the proposed systems with the previous state of the art should be carried out. In previous works, modified zeolites have proved to be able to adsorb and mineralize model dyes very efficiently under UV or also visible light, with very high removal efficiencies too. Some examples are: https://doi.org/10.1021/ie060641o; https://doi.org/10.3390/ijms23031730;https://doi.org/10.3390/catal7110344; https://doi.org/10.3390/ijms23031730. Can the Authors comment on this and discuss what is the advantage of this system with respect to the previous ones?
We thank the respected reviewer for this question. We subsequently discussed this topic (Lines 220-224). Our research aimed first to examine the adsorption and photocatalytic abilities of the original synthetic zeolite samples to remove MB and RB dyes. In further investigation, the mentioned zeolites can be modified to improve photocatalytic capabilities, which will be the subject of our further research.
There is no clear definition of “degree of mineralization” and how it is computed in Section 3. In the introduction it seems that mineralization is a different parameter than the COD. A clearer explanation for this is welcome.
Thank you very much for noticing this. As the reviewer requested, we added a more straightforward explanation of the terms mineralization and parameter COD and how they are connected. Therefore text in the introduction and chapter on Analytical procedures was clarified (Lines 105-106; 424-430).
Fig. 1 and others. The shape of the solid line passing through the points of the Freundlich isotherms is quite unusual. Must the line pass through the points? Can the line simply interpolate the points? Which is the error bar for each point?
In our opinion, plateaus (from which the number of adsorbed molecules was read) are visible if the curves pass through the points. And we hope that the respected reviewer will agree with our style of presenting the results. Adsorption parameters were determined from graphs of functional dependences ln x/m (ie. ln qe) vs. ln γe, where the line interpolates the points (we subsequently added them to the Supplementary materials). We added error bars for each point.
Line 132. The different behaviour of NH4BETA and NH4ZSM-5 is attributed to the presence of acid active sites of various strength. Are there also different site densities across the various zeolites? The number of sites can have an influence on the number of molecules accommodated within the pores.
By this, we thought that the different behavior of NH4BETA and NH4ZSM-5 zeolites with NaY zeolite is attributed to a higher SiO2/Al2O3 ratio, which indicates the presence of a more significant number of acidic active sites (different strengths and different densities) that favor the adsorption of these basic dyes.
Line 145. The different degree of dye removal is explained by the larger specific surface area of NH4BETA zeolite. What about the specific pore volume of these solids (micro + mesopores)? have the Authors data about this?
Thank you very much for the suggestion. We don't have the opportunity to measure a specific pore volume. In our discussion, we focused on the data of the specific surface because it includes the external and internal surfaces of all channels, cavities, and pores.
Fig. 9 and 10. Data for blank tests (in the absence of zeolite catalyst) should be added, to evaluate the contribution of self-photodegradation (if any).
Thank you for the suggestion. We have added a graph Figure S13 (in the Supplementary material) with data on the degree of dye photodegradation in the absence of zeolite. Therefore, additional text was added to the Photodegradation and mineralization chapter (Lines 232-236).
Line 234. The Authors invoke the role by adsorbed oxygen molecules. Did the Authors try to perform the tests under inert atmosphere?
Based on many papers read and previous experience, we decided to conduct experiments in the presence of oxygen. We believe that the work is precise and aims more for practical application than removing the investigated compounds in an inert atmosphere. Although we understand why the reviewer asked this question, we will examine the inert atmosphere's influence on the process's efficiency in future research.
Table 1. Which is the uncertainty of the pH values reported? I guess the pH value at 0 min for NH4ZSM-5, even if at different temperatures, should be very similar. On the contrary, two quite different values were obtained: 7.12 and 6.36. If so, are the rest of the variations significant enough? In some cases, pH increase or diminution is only 0.1.
The reviewer is correct. Sorry for this technical error for all pH values at 293 K for NH4BETA and NH4ZSM-5. We have corrected the error in the table. Also, we presented our results in one decimal place.
Section 2.3. It is not clear how relevant is the first part of the computational analysis. In all examples, Cl counterion is placed in a well-determined position. However, during the adsorption process in real experiments, Cl- anion is freely dissolved in the aqueous medium and does not likely enter the zeolite channels, as it would be repelled by the mainly anionic framework of the BEA, MFI or FAU, caused by Al sites. Are the reddish/brown areas in Fig. 14 relevant to explain the real behaviour in the experimental tests? In addition, the Authors mention a “sensitivity toward electrophilic attacks”. In the adsorption process, there should be a prevalent electrostatic attraction between the positively-charged dye and the negatively-charged zeolite cavities. What is the electrophilic species that is responsible for the attack? Please, explain.
Thank you very much for raising this question, as we were also thinking about this topic when starting with molecular modeling in this work. We agree that the is freely dissolved in the aqueous medium and that the chances it enters the zeolite channels are low. However, although it is a more challenging approach, we decided to consider dyes in the presence of . Although are not likely to be adsorbed by the zeolites, they are still present in the solution and might influence the charge distribution within dye molecules. We also searched the literature and found the following study https://doi.org/10.3390/molecules26133780 where the influence of was taken into account during the adsorption of methylene blue by TiO2 surface.
Regarding the results presented in Figure 14, we believe that the results related to the charge distribution are relevant because DFT calculations indicate a vast difference in the maximal MEP values between methylene blue and rhodamine B. We agree with you that during adsorption, electrostatic interactions are predominant. However, sensitivity towards electrophilic attacks might be important for the degradation process, indicating where this process might be initiated. At this stage, it is challenging to identify electrophilic species that might be responsible for the electrophilic attacks of methylene blue and rhodamine B dyes.

Reviewer 3 Report
I carefully reviewed this manuscript. The aim of this study is interesting. If this manuscript is modified or revised, there is a possibility that this manuscript is accepted. To attain the acceptance, the authors have to rewrite.
Although I can understand that the authors would like to emphasize the availability of the used zeolite samples, the organization of manuscript should be revised or modified. In addition, experimental results and analysis are week. Although I don’t know the acceptance rate of this journal “molecules”, this manuscript is required to modified or revised for acceptance.
Abstract
P1, L17: Change to the past tense. “we have studied” >>> “we studied”
P1. L19: The term of “zeolites” is used. In this case, which is correct, “untreated zeolites” or “zeolite-based composite”?
P1, L20-26: In abstract, the description on procedure is required. The main results (and related discussions) should be described (added).
P1, L28: I can’t understand the term “in the RB dye case”. I imagine “for (removal) of RB (dye)”.
Introduction
P1, L35: “Water quality is deteriorating yearly” changes to the perfect tense, for example “Water quality has deteriorated yearly”.
P1, L41: “small concentrations” I think that “low concentrations” is more frequently used and better.
P1, L41: “dyes' presence” I think that the possessive case like this should be used and “the presence of dyes” is better.
P1, L43: “the kidneys, liver” one is singular from and the other is plural form. “(the) kidneys, livers” is better.
P1, L44: I can’t understand the pronoun ‘they” points at which?
P2, L48: I think that the term “They are widely used” is changed to “They have been widely used”.
P2, L77: “most often” >>>”often” or”frequently”
P2, L85: I can’t understand the meaning of the term “the first”. This sentence should be modified.
P2, L95-102: At the last section in Introduction, the aim and content of this manuscript is shown. In this case, the possessive form is not used.
“3. Materials and Methods” section is changed to “2. Materials and Methods” section.
P12, L335: The authors selected MB and RB as target dye. If you compare the removal of them, it is better that the dye concentration is unified in molar concentration. However, they are used in weight concentration, mg/dm3.
P12, L337: How was the used zeolite samples prepared (synthesized)? At least, short explanation (description) on the preparation or synthesis procedure is required. It is lack of information for readers.
P12, L344: “FT-IR spectrum” is changed to “FT-IR spectra”.
P12, L349: “at temperatures of 283 K and 293 K” is changed to “at 283 K and 293 K”. What is the reason to select these two temperatures? The difference is only 10ÌŠC. What is considerable effective difference of experimental results?
P12, L352-353: The content in Supplementary materials is transferred to the main text.
P12, L361: The content in Supplementary materials is transferred to the main text.
2. Results and Discussion
P3, Figure 1: “Freundlich adsorption isotherm” is changed to “Freundlich adsorption isotherm curves”. The curves should be smoothly drawn. The line is not needed to pass though the all plots. Same as other figures.
Did dye adsorption attain the equilibrium only for 180 min? If possible, the time source of typical dye adsorption (or dye concentration) is shown in supplementary materials? It is important.
P3: Did the authors analyze the adsorption kinetics by Langmuir isotherm? Why the author determined these adsorption processes as physical adsorption from these results? I think evidence is week. More evidence is required.
Figures 1-6: Some of the data are more or less varied. Are data averaged?
Figures 7 and 8: The caption of x axis is not shown. Some of the adsorption data attained the removal % of 100. Dye components in the solutions are completely removed by adsorption. This means that the dye concentration is too diluted. I think the results at higher dye concentrations are required.
Figure 9: the term “radiation” is overlapped in title.
On photodegradation: Does the degradation of dyes occur in the presence of zeolite samples? If so, the degree of degradation in the absence of (without) zeolite is lower than those in the presence of zeolite samples. I think that these data are required.
Although I can understand that the authors would like to emphasize (described) good availability of the used zeolite samples, the experimental results and analysis are week (poor). More results are required to explain the availability of the used zeolite samples.
References
Some of the journal names are not abbreviated. For example, refs. 3, 4, 7, 8, 10, 11, 22, 29, 31, 32-34, 36, 37, 38-42, 48, 50, 51, and 56.
Author Response
Journal Molecules (ISSN 1420-3049)
Manuscript ID molecules-1897082
Title: Zeolite-based composite for removal of dyes from water by adsorption and photocatalysis
Special Issue Modeling Adsorption Properties of Molecular and Nanostructured Systems for Environmental Applications
The authors thank the reviewer for their constructive comments and recommendations. We have taken the comments on board to improve and clarify the Manuscript. Please find below a detailed point-by-point response to all comments.
The reviewers’ comments are typed in black. The responses to reviewers are tagged blue. The changes in the text of the Manuscript are typed in red. The line numbers in responses to reviewers refer to the corrected text.
Response Reviewer 3:
-I carefully reviewed this manuscript. The aim of this study is interesting. If this manuscript is modified or revised, there is a possibility that this manuscript is accepted. To attain the acceptance, the authors have to rewrite.
Although I can understand that the authors would like to emphasize the availability of the used zeolite samples, the organization of manuscript should be revised or modified. In addition, experimental results and analysis are week. Although I don’t know the acceptance rate of this journal “molecules”, this manuscript is required to modified or revised for acceptance.
We thank the reviewer for comments and the opportunity to improve our manuscript.
Abstract
P1, L17: Change to the past tense. “we have studied” >>> “we studied”
Thanks for the suggestion. We corrected the abstract and did not use the possessive form, so we changed “we have studied” to “This study investigated” (Line 17).
P1. L19: The term of “zeolites” is used. In this case, which is correct, “untreated zeolites” or “zeolite-based composite”?
In this manuscript, we used original samples of synthetic zeolites (Zeolyst International, USA). In accordance with that, we changed the term “zeolites-based composite” in the title and abstract (Line 17). Thank you for pointing out this important item.
P1, L20-26: In abstract, the description on procedure is required. The main results (and related discussions) should be described (added).
We corrected the abstract and added the main results (Lines 23, 29-31).
P1, L28: I can’t understand the term “in the RB dye case”. I imagine “for (removal) of RB (dye)”.
Thanks for the suggestion. We reworded the sentence (Line 32).
Introduction
P1, L35: “Water quality is deteriorating yearly” changes to the perfect tense, for example “Water quality has deteriorated yearly”.
Thank you for the suggestion. We changed the sentence to the perfect tense (Line 39).
P1, L41: “small concentrations” I think that “low concentrations” is more frequently used and better.
Thank you very much for this correction. We agree that "low concentrations" is a much better term, and we used it (Line 45).
P1, L41: “dyes' presence” I think that the possessive case like this should be used and “the presence of dyes” is better.
We changed it to "the presence of dyes", which also sounds better to us (Line 45).
P1, L43: “the kidneys, liver” one is singular from and the other is plural form. “(the) kidneys, livers” is better.
Thanks for the suggestion. We corrected the plural (Line 47).
P1, L44: I can’t understand the pronoun ‘they” points at which?
To clarify the sentence, instead of „they“ we used „also dyes“ (Line 48).
P2, L48: I think that the term “They are widely used” is changed to “They have been widely used”.
Thank you, we have changed the sentence as you suggested (Line 52).
P2, L77: “most often” >>>”often” or”frequently”
We changed it to „often“ (Line 83).
P2, L85: I can’t understand the meaning of the term “the first”. This sentence should be modified.
We have changed this sentence and believe it is written more clearly now (Lines 91-94).
P2, L95-102: At the last section in Introduction, the aim and content of this manuscript is shown. In this case, the possessive form is not used.
Thanks for noticing. We rephrased the sentence.
“3. Materials and Methods” section is changed to “2. Materials and Methods” section.
Although we usually write “Materials and Methods” first and then “Results and Discussion”, here we put materials and methods in the third chapter because the journal Molecules suggests this in the template.
P12, L335: The authors selected MB and RB as target dye. If you compare the removal of them, it is better that the dye concentration is unified in molar concentration. However, they are used in weight concentration, mg/dm3.
We found that the literature expresses dye concentrations in mass or molar concentrations. We would be happy to accept your suggestion. Still, if we now convert to molar concentration, we would get concentration values for RB and MB that could not be compared due to different numerical values of their molar masses.
P12, L337: How was the used zeolite samples prepared (synthesized)? At least, short explanation (description) on the preparation or synthesis procedure is required. It is lack of information for readers.
Synthetic zeolites which were used during the work were products of the company Zeolyst International, USA: NH4BETA (from the BEA group, mark CP 814E), NH4ZSM-5 (from the MFI group, mark CBV 3024E), and NaY (from the FAU group, mark CBV 100), which is highlighted in chapter Chemicals and solutions.
P12, L344: “FT-IR spectrum” is changed to “FT-IR spectra”.
We changed it to “FT-IR spectra” (Line 380 and Figures S16-S18).
P12, L349: “at temperatures of 283 K and 293 K” is changed to “at 283 K and 293 K”. What is the reason to select these two temperatures? The difference is only 10ÌŠC. What is considerable effective difference of experimental results?
Thank you very much for this question. Water temperatures in most of Europe area usually range from 283 K to 293 K throughout the year. We strived for as realistic conditions as possible, which is why we chose these two temperatures.
We have also changed to “at 283 K and 293 K” as per your recommendation (Line 385).
P12, L352-353: The content in Supplementary materials is transferred to the main text.
We have transferred the content from Supplementary materials to the main text (Lines 389-398).
P12, L361: The content in Supplementary materials is transferred to the main text.
We have transferred the content from Supplementary materials to the main text (Lines 404-410).
- Results and Discussion
P3, Figure 1: “Freundlich adsorption isotherm” is changed to “Freundlich adsorption isotherm curves”. The curves should be smoothly drawn. The line is not needed to pass though the all plots. Same as other figures.
We changed the name to "Freundlich adsorption isotherm curves". The line in the Freundlich isotherms passes through all points to clearly define and observe the plateaus from which we read the number of adsorbed molecules. We subsequently added graphs of the functional dependence ln x/m (i.e. ln qe) vs. ln γe in the Supplementary materials where the line interpolates the points. Adsorption parameters (k, n) were determined from these graphs.
Did dye adsorption attain the equilibrium only for 180 min? If possible, the time source of typical dye adsorption (or dye concentration) is shown in supplementary materials? It is important.
Your question is quite logical. We will try to explain. In some systems (zeolite dye), the equilibrium state was reached in 180 min (e.g., in the case of the RB-NH4ZSM-5 system at 293 K, RB-NaY at 283 K). That we would be able to compare all the obtained results, we monitored the adsorption consistently in the same time interval.
P3: Did the authors analyze the adsorption kinetics by Langmuir isotherm? Why the author determined these adsorption processes as physical adsorption from these results? I think evidence is week. More evidence is required.
Thank you very much for this question. We also observed the adsorption using the Langmuir isotherm. However, the Freundlich isotherm model proved to be much better. The correlation coefficient of the Freundlich model (R2) for all systems was significantly higher than the correlation coefficient of the Langmuir model. According to the Langmuir isotherm model, the highest R2 was for the RB-NH4ZSM-5 system (0.9568 at 283 K; 0.9577 at 293 K) and MB-NH4BETA system (0.9643 at 283 K). However, the stated values are smaller than the Freundlich models’ correlation coefficients. According to the Freundlich isotherm model, R2 was 0.9939 for RB-NH4ZSM-5 (at 283 K) and 0.9986 (at 293 K) and for MB-NH4BETA system was 0.9940 (at 283 K). The high values of the correlation coefficients R2 in the Freundlich model show that this model is much more suitable for the adsorption process of MB and RB dyes. The Freundlich isotherm model indicates multilayer physical adsorption, and the calculated adsorption parameters are given in Tables S1 and S2.
Figures 1-6: Some of the data are more or less varied. Are data averaged?
Each measurement was repeated three or more times, and the average of the date was calculated.
Figures 7 and 8: The caption of x axis is not shown. Some of the adsorption data attained the removal % of 100. Dye components in the solutions are completely removed by adsorption. This means that the dye concentration is too diluted. I think the results at higher dye concentrations are required.
The x-axis indicates the initial dye concentrations (γo), ranging from 0.5 to 5 mg/dm3. We chose this range of concentrations because we aimed for as realistic conditions as possible. Some studies show that dyes in natural waters are sometimes present at a concentration of 1.56 mg/dm3, and the dye in a clean river becomes visible already at a concentration of 0.005 mg/dm3.
Figure 9: the term “radiation” is overlapped in title.
Thank you notice. We deleted one “radiation” after “180 min” in Figures 9 and 10.
On photodegradation: Does the degradation of dyes occur in the presence of zeolite samples? If so, the degree of degradation in the absence of (without) zeolite is lower than those in the presence of zeolite samples. I think that these data are required.
Thank you very much. We added a graph (in the Supplementary material) with data on the degree of dye photodegradation in the absence of zeolite. Degradation of dyes showed less efficiency in the absence of zeolites. We aded apropiate text in Lines 232-236.
Although I can understand that the authors would like to emphasize (described) good availability of the used zeolite samples, the experimental results and analysis are week (poor). More results are required to explain the availability of the used zeolite samples.
We hope that the respected Reviewer will be satisfactory how we improved our manuscript, and after this revision, it will be recommended for acceptance.
References
Some of the journal names are not abbreviated. For example, refs. 3, 4, 7, 8, 10, 11, 22, 29, 31, 32-34, 36, 37, 38-42, 48, 50, 51, and 56.
We are very grateful for the detailed reading and observed errors that we have corrected.

Round 2
Reviewer 1 Report
The authors made a thorough revision of the manuscript according to the suggestions given and answered all the questions.
The manuscript is suitable for publication in "Molecules" in the present form.
Author Response
The authors thank the reviewer for their positive opinion and recommendation of the Manuscript for publishing.
Best regards,
Sanja Armaković

Reviewer 3 Report
I think this manuscript will be published in this journal.
Before publication, some graphs should be drawn again.
The curve should be drawn smoothly, not passed through the plots For Figures 1-6.
Author Response
The authors thank the reviewer for their recommendations. We again drew graphs (Figs. 1-6) and added one sentence as an explanation (Line 113).
The new changes in the text of the Manuscript are typed in green.
We hope the manuscript now meets the criteria for publication in Molecules Journal.
Best regards,
Sanja Armaković
